# Factors influencing adoption of oral health promotion by antenatal care providers in Moyo district, North-Western Uganda

**Patrick Madrama Lulu** [1]*, **Miisa Nanyingi** [2]

**1** Faculty of Allied Health Sciences, Clarke International University, Kampala, Uganda, **2** Faculty of Health Sciences, Uganda Martyrs University, Kampala, Uganda

* lulu.madrama@gmail.com

## Abstract

### Background

Oral health promotion (OHP) during pregnancy is an important global public health and basic human right issue related to quality of life. Several statements and guidelines have been published emphasizing the need for improved oral health care of pregnant mothers, prenatal care providers have missed this critical opportunity. In this study, we assessed factors influencing adoption of oral health promotion by antenatal care providers.

### Materials and methods

A descriptive cross-sectional study design that employed both quantitative and qualitative data collection methods and analysis. 152 samples determined using Yamane's 1967 and stratified sampling technique was used. Three FGDs and six KI interviews were held. Univariate, bivariate and multivariate analyses were done using SPSS (20.0) and ATLAS Ti for qualitative analysis.

### Results

Adoption of OHP was low 28% (42). Factors influencing adoption were age of respondents (OR = 0.066, 95%CI = 0.009–0.465, p = 0.006*), level of care of health facility (OR = 0.050, 95%CI = 0.008–0.322, p = 0.002*), good understanding between dentists and ANC providers (OR = 0.283, 95%CI = 0.084–0.958, p = 0.042*), availability of practice guideline for OHP in ANC (OR = 0.323, 95%CI = 0.108–0.958, p = 0.043*), number of years at work (p = 0.084), being knowledgeable (OR = 2.143, 95%CI = 0.864–5.311, p = 0.100), having skills to advance OHP(OR = 0.734, 95%CI = 0.272–1.984, p = 0.542), Management being good at influencing new practices (OR = 00.477.734, 95%CI = 0.227–2.000, p = 0.477). More emphasis on national and local of oral health issues, continuous staff training on oral health, dissemination of National oral health policy (NOHP) were some of key issues that emerged from the qualitative results.

**Data Availability Statement:** All relevant data are within the paper.

**Funding:** The authors received no specific funding for this work.

**Competing interests:** The authors have declared that no competing interests exist.

**Abbreviations:** ANC, Antenatal care; aOR, Adjusted Odds Ratio; CI, Confidence Interval; MoH, Ministry of Health; FDG, Focus Group Discussion; KI, Key Informants; uOR, Unadjusted Odds Ratio; UMU, Uganda Martyrs University; HSSIP, Health Sector Strategic and Investment Plan; WHO, World Health Organization.

## Conclusion

Adoption of OHP was low. This was attributed to age, number of years spent at work, level of health facility, having good understanding between dentists and ANC providers, availability of practice guidelines, dissemination of National oral health policy, continuous staff training. We recommend the current NOHP to be reviewed, develop prenatal OHC guidelines, enhance the capacity of ANC providers through training, collaboration with dentists and launch official adoption of OHP.

## Background

Oral disease is one of the most important public health issues in the world, with significant socio-economic impacts [1]. Among pregnant women, common oral health diseases are periodontal diseases with a high prevalence rate [2]. Oral health is a basic human right fundamental to people's quality of life [3], achieving oral health goals would require health systems being supported by enabling actions in education, improved nutrition and hygiene [4]. Antenatal care is the care provided by skilled healthcare professionals to pregnant women and adolescent girls to ensure the best health conditions for both mother and baby during pregnancy and its components include; risk identification, prevention and management of pregnancy-related or concurrent diseases including health education and health promotion [5]. Pregnancy has an impact on oral related quality of life [6]. Pregnancy provides an opportunity for women to initiate new healthy behaviours, including oral health practices, dental and prenatal care providers have missed this critical opportunity to promote oral health [7]. Some of the missed opportunities for oral health promotion include; oral health education, oral health examination and referral for dental services when mothers are receiving antenatal care.

Globally, several statements and guidelines have been published emphasizing the need for improved oral health care of pregnant mothers, because, hormonal changes during pregnancy combined with neglected oral hygiene tend to increase the incidence of oral diseases including gingivitis [8]. The United States of America, Department of Health and Human Services highlighted the need for partnerships between dentists and other health professionals, including nurses and midwives and these recommendations have resulted in the development of evidence-based practice guidelines for oral health care during pregnancy and early childhood for all health professionals including prenatal care providers [9].

In low and middle-income countries, the prevalence of oral diseases continue to increase, with tooth decay rapidly increasing among adults and there will be a huge burden of this health problem in the future if sustainable programs are not put in place [10]. The good oral health of a pregnant woman should be considered to be of utmost importance for dental practitioners and antenatal care providers [11]. Integration of Oral health into primary care has been implemented in some health care systems to reduce the burden of oral health disease and to improve access to oral health care, especially for disadvantaged populations and communities [12]. Asia developed a strategy in which emphasis was put on the integration of oral health promotion and other disease prevention [13], similarly, countries in other regions had implemented preventive strategies to maintain the oral health among pregnant mothers [14].

Adopting an integrated approach to healthcare would achieve better outcome for patients with oral diseases, with several policy recommendations that have been made including the adoption of oral health in all policies, strengthening inter-professional collaboration, and

inclusion of oral health in the curriculum for all healthcare professionals [15]. In addition, the WHO global policy for the improvement of oral health showed that, the promotion of oral health is a cost-effective strategy to reducing the burden of oral diseases and maintaining oral health and quality of life. Oral Health services should focus on prevention and early diagnosis [16].

In Sub-Saharan Africa, there is an increased prevalence of dental diseases, resulting from increased consumption of sugars and inadequate exposure to fluorides [17]. In Mali, there was increased burden of periodontal diseases [18], while in Tanzania, there was increased burden of bleeding gum, dental pain, tooth decay and swollen gums [19]. Nigeria's National oral health policy indicated the need for oral health to be included as a component of health promotion and should be provided for during antenatal care services [20] and emphasis has been placed on integrating oral health promotion in maternal and child care among African Countries [21].

In Uganda, the prevalence of oral diseases among women was the highest, 42.4% compared to the general population [22], 67.0% of pregnant mothers having periodontal diseases [23], 86.0% postpartum mothers having plague deposits [24]. Uganda has emphasised that appropriate promotion of oral health requires integration of oral health policy elements, strategies and policies of all sectors to impact community health, including maternal and child health [25]. In Moyo district, there is limited data on the level and factors associated with adoption of Oral health promotion by antenatal care providers.

## Materials and methods

### Study population and settings

This cross sectional study on adoption of oral health promotion was done among antenatal care providers working in a general hospital, Health Centre IV and health centre IIIs within Moyo district between August 2017 to October 2017.

The inclusion criteria required that, for the ANC provider to participate in the study, they should be working at ANC units, those who had ever worked at the ANC unit, these included Nurses, Midwives, Clinical Officers and medical doctors within the selected health facilities from whom written informed consent was obtained to participate in the study. We excluded Nurses, Midwives, Clinical Officers, Gynaecologists, paediatricians, general medical doctors who had never worked at the ANC department before the study and ANC providers who had spent less than six months working at the time of this study, as they may not have practical and authentic responses to the study compared those who had ever worked at ANC departments for more than six months.

### Study variables

**Adoption of Oral Health Promotion** as a dependent variable was measured using 24 items of practices discussing importance of oral health, asking questions related to oral health practices, providing counselling, teaching on how to rinse mouth, teaching on tooth brushing, counselling on association between poor periodontal health and negative birth outcomes, advise on visiting dentist, provide information on the need for limiting sweat foods and drinks, foods and drinks that reduce once chances of developing oral diseases, need for food supplements like Calcium and vitamins, maintaining healthy body weight, tobacco use and alcohol consumption, educate about oral hygiene, advice pregnant mothers to chew sugarless gums after eating, rinse mouth with a teaspoonful baking soda in a cup of water after vomiting to neutralize acid, brush teeth twice a day using fluoride toothpaste, advise mothers to avoid sharing saliva and spoon, taught how to wipe baby's teeth when the first tooth erupts after feeding with

a soft cloth toothbrush, teach mothers to avoid putting their baby to bed with a bottle or Sippy cup containing anything other than water, take oral health history, check mouth for sign of bleeding, cavity or sign of infection, document findings, reassure about safety of oral health care, dental visit every after 6 months for related oral health care services by antenatal care providers, with a total mean sore of 18 (75%, SD 3.2) and a greater proportion of ANC providers had not adopted OHP during ANC.

**Knowledge** influencing adoption of oral health promotion: 11 items were used (dental diseases among pregnant mothers have effect on pregnancy, physiological changes during pregnancy predispose mothers to gum diseases, pregnancy accelerates existing dental problem, poor maternal oral health can contribute to early childhood teeth decay, women should receive preventive dental care during pregnancy, it safe to obtain dental radiographs in pregnant women, pregnant women should only receive emergency dental care, periodontal diseases is common among pregnant women, I am concerned about being sued if something goes wrong in a pregnancy, it will be easy to understand and adopt oral health promotion during ANC services, adoption of oral health promotion in ANC services will meet the oral health need of pregnant mothers), with a mean score of correct answers 7 (64%, SD 2.2).

For measuring attitude of the respondents, 6 items were used (whether it is necessary to adopt oral health promotion in ANC services, interested of participating in promoting oral health for pregnant mothers, asking pregnant women about oral health is outside routine ANC practice, pregnant women are comfortable with assessing oral health during normal antenatal check-ups, will stand out to advocate for adoption of oral health promotion in ANC services, I first want to see the results of adopting OHP in ANC services before I take part), with a mean score 4 (66%, SD 0.98).

**Capacity of Health facilities** to provide oral health promotion activities: 16 items were used (not easy to adopt OHP into antenatal care services because of limited number of staff, there is no time to promote oral health during ANC services, good understanding exist between dentists and other health workers, OHP for pregnant mothers require inter-professional collaboration, do not access web-sites that focuses on OHP, no practice guidelines on Oral Health care during pregnancy, facility do not have posters for oral health, ANC providers can recognize and offer appropriate dental care and referrals, no attention given to Oral health of pregnant Mothers, dental conditions for pregnant mothers are referred outside this health facility, most oral health services are not available, some oral health services are expensive for the pregnant mothers, never been supported to advance my carrier, been given skills to promote oral health, always been heard and recognised) and capacity was measured through binary scale (no or yes).

11 items used for strategies to influence adoption OHP (management good at influencing workers to implement new practices, have a common understanding among staff, necessary to involve staff members in formulating policies and review on adoption of OHP, need to develop oral health promotion guideline, develop healthy policy, my views are important in developing policy and guideline, interested in further information about dental care, received training on oral health, have the skills to advise pregnant women on oral health, important to design training package for oral health promotion, training on basic oral health assessment will be helpful for ANC providers) this was measure through binary scale (no or yes) and there were inadequate strategies to influencing oral health promotion by ANC providers.

## Study design

A descriptive cross-sectional study was used where both quantitative and qualitative methods of data collection and analysis were employed. A concurrent data collection approach was

used including information and the interpretation of overall results [26], this design was used because of the shorter data collection period and the need to provide a comprehensive analysis of the research problem.

152 samples were determined using Yamane's (1967) formula $\mathbf{n} = \frac{N}{1+N(e)^2}$, Where

n = sample size, N = Number of ANC providers that included Nurses, Midwives, Clinical Officers and Medical Doctors in the selected health facilities within Moyo district (245), e = Level of precision or sampling error (0.05) and a stratified sampling technique was used. Three FGDs and six KI interviews were held. Univariate, bivariate and multivariate analyses were done using SPSS (20.0) for quantitative analysis and ATLAS Ti for qualitative analysis.

## Data analysis

We analysed Quantitative data using Microsoft Excel 2010 and SPSS 20. Chi-square statistics were computed to check for statistically significant differences in the parameters between the dependent and independent variables (Univariate, bivariate and multivariate analysis were performed in SPSS version 20.0). Logistic regression was done to obtain strength of association between categorical dependent and independent variables and statistically significant items at this level were then analysed using multivariate logistic regression to obtain adjusted conclusions for the finding of the study. Qualitative data were transcribed verbatim, coded and organised in themes in ATLAS.ti. Triangulation was done to incorporate both the quantitative and qualitative data obtained from respondents.

## Ethical considerations

This study was conducted in accordance with the Helsinki declaration guideline, a waiver was granted by International Health Sciences University Research Ethics Committee (IHSU-REC) (IHSU-REC/004). Prior to the waiver, administrative clearance was done by Uganda Martyrs University Scientific Review Committee, a written consent was obtained from Moyo District Health Office (DHO), verbal consent from the medical superintendent of Moyo hospital, incharges of the Dental unit, Antenatal care department and various health facilities under study and finally written informed consent obtained from participants under study.

## Results

### Demographic characteristics of the respondents

Our study considered a total of 152 ANC providers, mean age 36, range 37 years, a standard deviation of 10.49, of which, 57(37.5%) were 37 years and above, 92(60.5%) females, 106 (69.9%) had worked between 1–10 years, 82(53.9%) working HC IIIs, 67(44.1%) Nurses and 82(53.9%) were Certificate holders. (See *Table 1*).

### Adoption of oral health promotion by ANC providers

42 (28%) of the respondents out of 152 had adopted oral health promotion for pregnant mothers. Although individual practices of ANC providers appeared to be low, the management and officer in charge of the MCH programme noted that, adoption or integration of oral health promotion in ANC would be vital for overcoming some of the challenges of poor oral health, its outcome on pregnancy and the child (See *Fig 1*).

*"There is need to adopt or integrate oral health promotion into ANC services to overcome some of the challenges created by the poor oral health of the pregnant mothers" (KI 1).*

**Table 1. Demographic characteristics of respondents.**

| Variables | Number (%) |
|---|---|
| | N = 152(%) |
| **Age** | |
| 22–26 | 29(19.1) |
| 27–31 | 33(21.7) |
| 32–36 | 33(21.7) |
| 37 & above | 57(37.5) |
| **Gender** | |
| Male | 60(39.5) |
| Female | 92(60.5) |
| **Number of years at work** | |
| 1–10 years | 106(69.7) |
| 11–20 years | 23(15.1) |
| 21–30 years | 8(5.3) |
| 31–40 years | 15(9.9) |
| **Level of Health Facility** | |
| Hospital | 40(26.3) |
| HCIV | 30(19.7) |
| HCIII | 82(54) |
| **Professional Cadre** | |
| Nurses | 67(44.1) |
| Midwives | 35(23.0) |
| Double trained Nurse/midwives | 24(15.8) |
| Clinical Officers | 20(13.2) |
| Medical doctors | 6(3.9) |
| **Professional Qualification** | |
| Certificate | 82(53.9) |
| Diploma | 60 (39.5) |
| Bachelor's Degree | 10(6.6) |

More emphasis supporting the need for adoption of Oral Health promotion by ANC providers showed that, adoption is likely to reduce heart diseases among pregnant mothers;

*"Of late there are increased cases of heart diseases. . . . . . originating from the dental issues as bacteria tend to move to the bloodstreams and later on move to the heart valves without knowing, adopting oral health promotion will have a possibility of reducing such challenges in the future" (KI 2).*

*"It will be cheaper adopting oral health promotion into ANC services compared to when it handled as a separate program" (FGD 1).*

## Factors influencing adoption of oral health promotion by ANC providers

In our multivariate analysis, age of respondents in years 22–26 (aOR = 0.066, 95%CI = 0.009–0.465, p = 0.006*), level of care of health facility (aOR = 0.050, 95%CI = 0.008–0.322, p = 0.002*), good understanding between dentists and ANC providers (aOR = 0.283, 95% CI = 0.084–0.958, p = 0.042*), availability of practice guideline for OHP in ANC (aOR = 0.323, 95%CI = 0.108–0.958, p = 0.043*), worked 1–10 years (p = 0.011*), were statistically significant

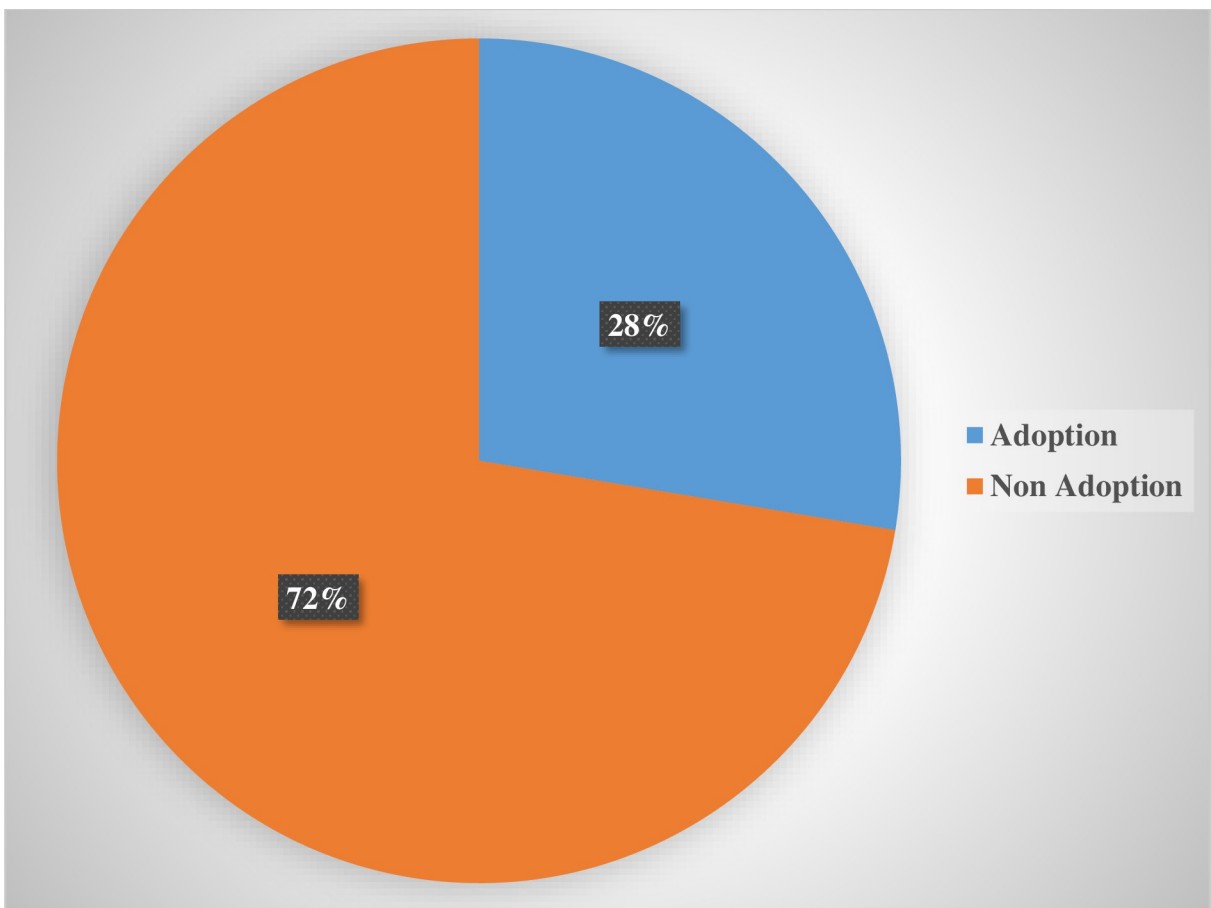

**Fig 1. Adoption of oral health promotion by ANC providers.**

and associated with adoption of oral health promotion by ANC providers. However, general number of years at work (aOR = 25.241, 95CI = 2.075–307.041, p = 0.084), being knowledgeable (aOR = 2.143, 95%CI = 0.864–5.311, p = 0.100), having skills to advance OHP in ANC (aOR = 0.734, 95%CI = 0.272–1.984, p = 0.542), Management good at influencing new practices (aOR = 00.477.734, 95%CI = 0.227–2.000, p = 0.477) were not statistically significantly associated with adoption oral health promotion by ANC providers (see *Table 2*).

> *"Top management at the national level is not good at influencing new policies and guidelines, there is no dissemination of the National Oral health policy, if dental health issues are not marketed by the authorities, no one will pay attention and it will continue to be neglected, and yet adoption will lead to a reduction on missed opportunities, maximizing resources–one staff offering all that is needed and improving service delivery" (KI 1).*

In our univariate analysis, female ANC providers were more likely to adopt oral health promotion compared to their male counterparts. Respondents aged between 22–26 years were less likely to adopt oral health promotion during prenatal care compared to those aged 37 years and above. Respondents who had been working for the last 21–30 years were 14 times more likely to adopt oral health promotion compared to response who had worked for 31–40 years (see *Table 2*).

**Table 2. Factors influencing adoption of oral health promotion by ANC providers in Moyo district, North-Western Uganda.**

| Variables | Adoption of Oral Health Promotion by ANC Providers | | Univariate Logistic Regression Analysis | | Multivariate Logistic Regression Analysis | |
|---|---|---|---|---|---|---|
| | Adoption (%) | Non-Adoption (%) | UOR (95%CI) | P-value | AOR (95%CI) | P-value |
| **Age in years** | | | | | | |
| 22–26 | 3 | 26 | 0.184(0.050–0.679) | 0.011* | 0.066(0.009–0.465) | **0.006*** |
| 27–31 | 8 | 25 | 0.509(0.195–1.327) | 0.167 | 0.148(0.029–0.741) | **0.020*** |
| 32–36 | 2 | 24 | 0.597(0.235–1.517) | 0.278 | 0.130(0.026–0.642) | 0.084 |
| 37 & above | 22 | 35 | 1 | | 1 | |
| **Number of years at work** | | | | | | |
| 1–10 years | 1 | 14 | 5.273(0.663–41.920) | 0.116 | 25.241(2.075–307.041) | **0.011*** |
| 11–20 years | 29 | 17 | 7.467(0.825–67.573) | 0.074 | 8.205(0.749–89.852) | 0.085 |
| 21–30 years | 8 | 15 | 14.000(1.200–163.367) | 0.035* | 12.064(0.775–187.885) | 0.075 |
| 31–40 years | 4 | 4 | 1 | | 1 | |
| **Level of Care** | | | | | | |
| Hospital | 2 | 28 | 0.827(0.366–1.869) | 0.647 | 0.317(0.097–1.034) | 0.057* |
| HC IV | 12 | 28 | 0.138(0.031–0.621) | 0.010* | 0.050(0.008–0.322) | **0.002*** |
| HCIII | 28 | 54 | 1 | | 1 | |
| **Knowledge** | | | | | | |
| Knowledgeable | 7 | 49 | 4.016(1.642–9.823) | 0.002* | 2.143(0.864–5.311) | 0.1000 |
| Not Knowledgeable | 35 | 61 | 1 | | 1 | |
| **Have skills to advance OHP in ANC** | | | | | | |
| No | 12 | 54 | 0.415(0.193–0.893) | 0.024* | 0.734(0.272–1.984) | 0.542 |
| Yes | 30 | 56 | 1 | | 1 | |
| **Management is good at influencing new practices** | | | | | | |
| No | 7 | 44 | 0.300(0.122–0.735) | 0.008* | 0.477(0.227–2.000) | 0.477 |
| Yes | 35 | 66 | 1 | | 1 | |
| **A good understanding exists between dentists and ANC providers** | | | | | | |
| No | 6 | 47 | 0.223(0.087–0.674) | 0.002* | 0.283(0.084–0.958) | **0.042*** |
| Yes | 36 | 63 | 1 | | 1 | |
| **Practice guidelines for OHP in ANC are available** | | | | | | |
| No | 28 | 54 | 0.482(0.229–1.013) | 0.054* | 0.323(0.108–0.958) | **0.043*** |
| Yes | 14 | 56 | 1 | | 1 | |

In terms of the level of health facilities, respondents working at the health centre IV were less likely to adopt oral health promotion compared to those working at HC IIIs. Knowledgeable respondents were four times more likely to adopt oral health promotion compared to those that were not knowledgeable and respondents who had no skills to advance OHP in ANC were less likely to adopt oral health promotion compared to respondents who had skills in advancing OHP for prenatal care.

Respondents who said that management was good at influencing new practices, were more likely to adopt oral health promotion compared to those who said that management was not good at influencing new practices. Having no good understanding between dentists and ANC providers less likely contributed to the adoption of OHP, and having no practice guidelines for OHP by ANC providers less likely led to the adoption of oral health promotion (See *Table 2*).

*"Even if the staff members make recommendations for some dental items, the oral health department issues are always pushed aside and not taken care of" (KI 3).*

*The whole district only offers extraction within government health facilities, other services are in a private dental clinic (KI 4).*

*We do not have oral health policies for ANC and there is no local guideline that is put in place yet. We do not have the National Oral Health policy even in our dental department and I have never seen it before" (KI 2).*

*There is a need for training of ANC providers on oral health promotion for pregnant mothers (FGD 1).*

The ANC providers were also asked to raise issues that should be included in the oral health promotion guideline. More than half 95 (62.9%) of the respondents mentioned almost all responses mentioned prevention of oral diseases, this was followed by oral health education 93 (61.6%), ways of improving access to oral healthcare 91(60.3%) and assessment of pregnant women's oral health 88 (58.3%) (See *Fig 2*).

*In line with participants' knowledge on* the kind of preventive oral care pregnant women should receive; *the majority 124 (89.9%) of the respondents mentioned how to maintain oral hygiene, 57(41.3%) of the ANC providers mentioned education on nutrition, less than half 45 (32.6%) mentioned oral health assessment and 29(21.0%) of the ANC providers mentioned referral, only 2(1.4%) mentioned other preventive care services as the kind of preventive services pregnant women should receive* (See **Fig 3**).

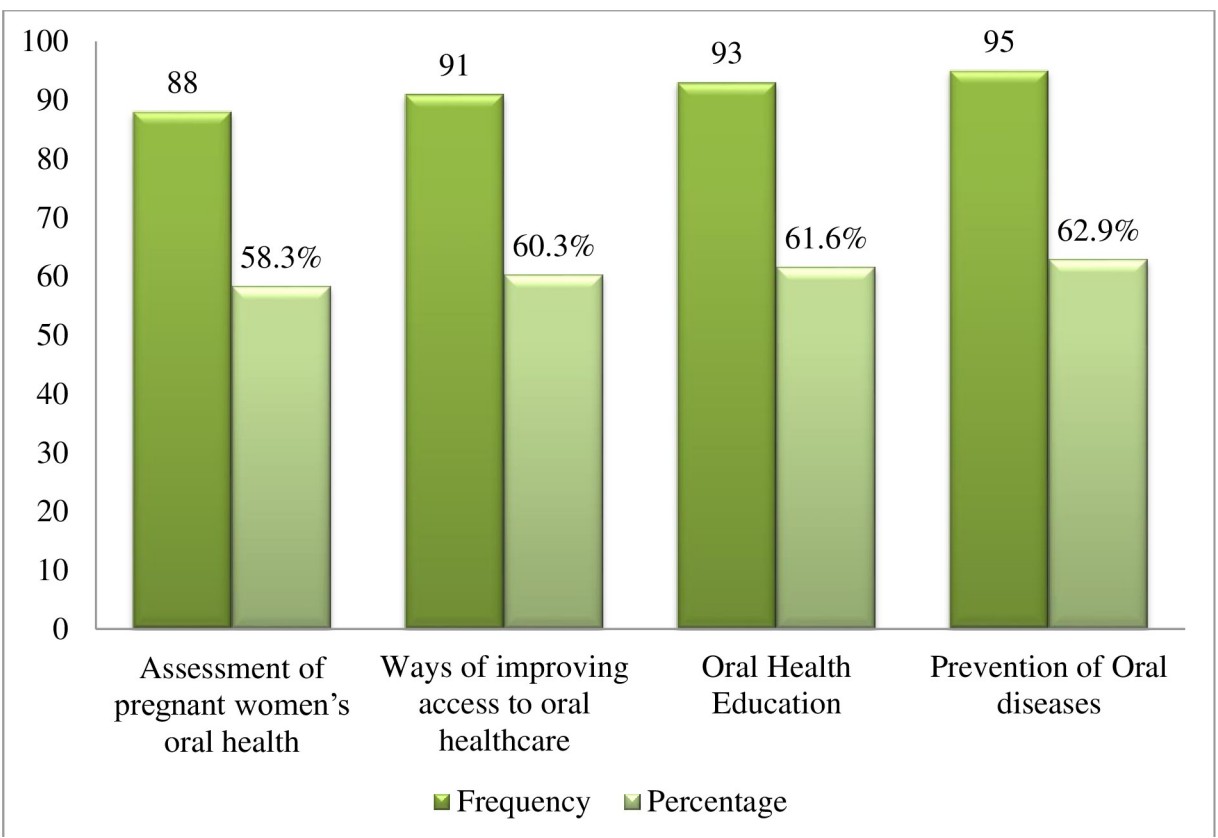

**Fig 2. Issues to be included in oral health promotion guideline during ANC.**

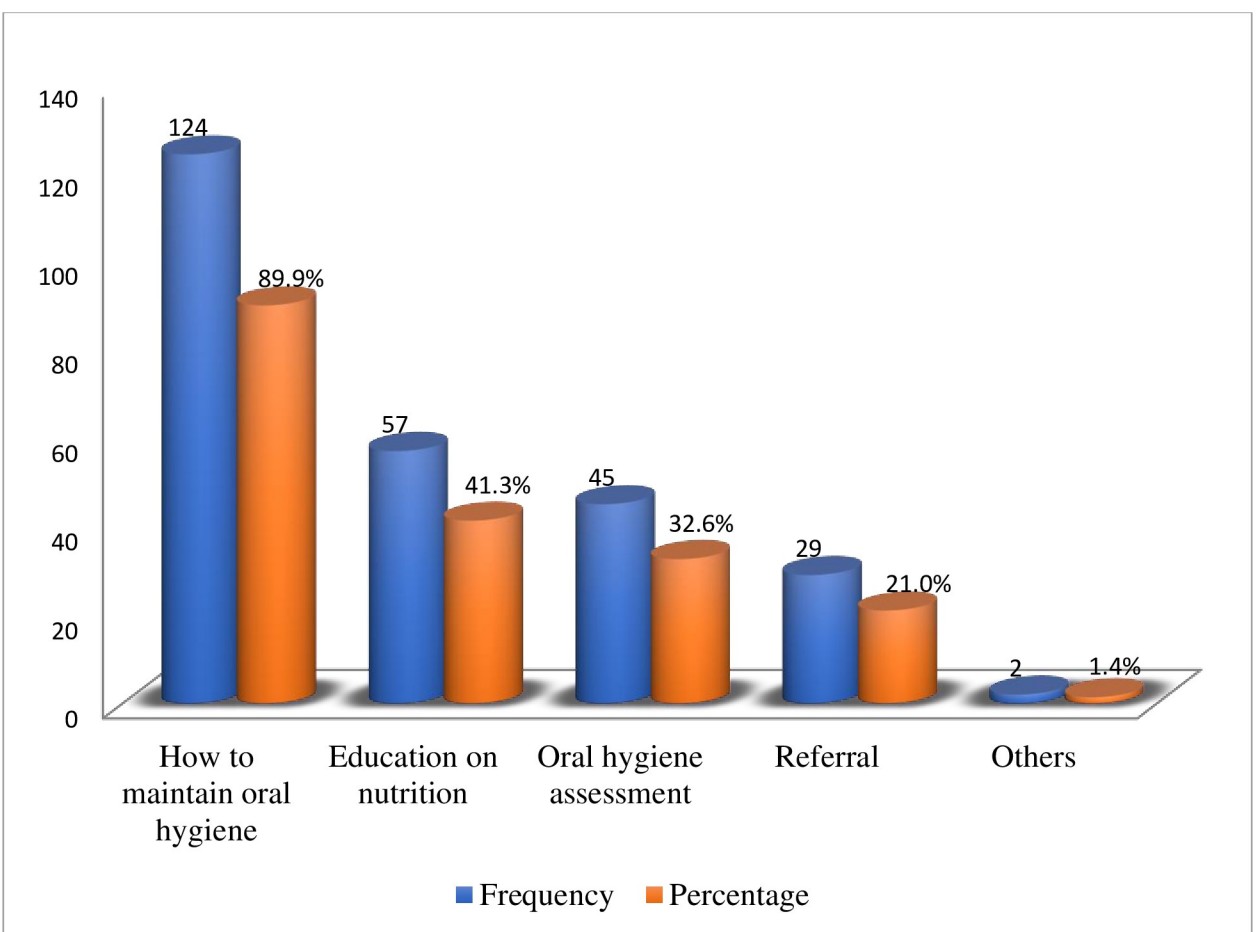

**Fig 3. Knowledge on type of preventive oral care pregnant women should receive.**

In line with the knowledge of respondents on the safety of obtaining dental radiographs during pregnancy, the majority 125 (82.2%) of the respondents were not aware that it is safe to obtain dental radiographs in pregnant women while, only 26 (17.1%) of the respondents said, obtaining dental radiograph in pregnant women is safe (See *Fig 4*).

## Discussion

### Adoption of oral health promotion by respondents

In this study, adoption of oral health promotion by ANC providers was low 42 (28%) of 152. Similarly, a study conducted in Australia among antenatal care providers showed that, very few ANC providers were involved in practices that promote oral health among pregnant women [27]. Adoption of oral health by ANC providers would play a significant role not only in the lives of pregnant mothers and also in their babies [28]. Maternal oral health behaviour could be shaped during antenatal care [29], promotion of oral health among pregnant women by Nurses would be feasible and need to be incorporated into the first antenatal booking visit and also recognized that oral health promotion was within their scope of practice [30].

Respondents aged between 22–26 years, 27–31 years, and 32–36 years were less likely to adopt oral health promotion during prenatal care compared to those aged 37 years and above. There was a statistically significant relationship between age and adoption of oral health

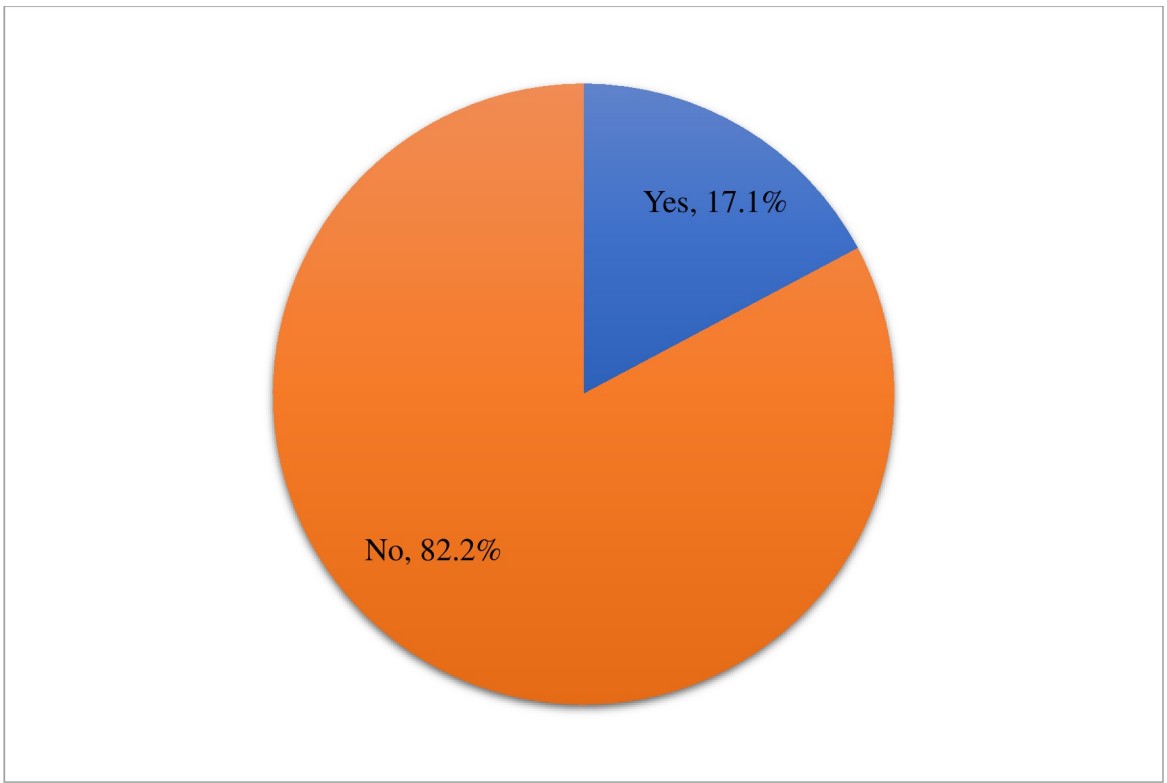

**Fig 4. Awareness on safety of obtaining dental radiograph during pregnancy.**

promotion. This may be due to experiences gained over the years about oral health. Contrary to the findings of our study, another study noted that, age is significantly associated with work-ability [31] and job satisfaction among Nurses [32].

Moreover, the number of years one spent at work improves occupational competence, consists of expertise, skills, and knowledge acquired through experience, training, and education, people's work are influenced by previous work experiences [33]. Our study noted that, respondents who had been working for the last 21–30 years were 14 times more likely to adopt oral health promotion compared to response who had for 31–40 years, who were 6 times more likely compared to those for 27–31 years, and 8 times more likely to those who had for 22–26 years respectively. Similarly [27], noted in their study that, there was an association between years spent at work and adoption of oral health promotion, Nurses 'experiences are closely related to the occurrence of adverse events [34]. However, Positive work environments are important in achieving patient safety, quality care and favourable patient outcomes [35].

Similarly, experience give Nurses' exposure to different patient or health-related conditions, including different clinical and preventive scenarios that would eventually contribute to increased knowledge, critical thinking and technical skills [36]. In general, job performance among nurses is motivated by years of experience [37]. However, it has also been noted that age and length of time in service within ANC do not influence knowledge of oral healthcare for prenatal mothers [38].

In terms of the level of the health facility where respondents were working, our study found that those working at the district general hospital were less likely to adopt oral health promotion compared to respondents working at HC IIIs. However, working at a general hospital contributed to 82.7% adoption compared to working at HCIV which only contributed to

13.8% adoption of oral health promotion by ANC providers. This may be because most of the care of HCIIIs in the district are offered by generally Nurses, Midwives and Clinical officers, ANC at this level of facility handles all aspects of preventive services including oral health education and promotion. The general hospital and HCIVs may be overwhelmed by several conditions because of referrals from lover facilities.

In our study, less than half 56(36.8%) of the respondents were knowledgeable about OHP in ANC and those knowledgeable were four times more likely to adopt oral health promotion compared to those that were not knowledgeable. Similarly, Nurses and Midwives had insufficient knowledge of oral healthcare [38]. Respondents in our study were not knowledgeable about the fact that several physiological changes take place among pregnant women and these have an effect on oral health, pregnancy and brings about several changes in the oral cavity [6]. Our finding was similar to the study in Australia which noted that ANC providers had limited knowledge on prenatal oral healthcare [27].

In terms of knowledge on the safety of dental procedures during pregnancy, our study found that ANC providers were not knowledgeable about the safety of local anaesthetics, teeth extraction, scaling and Root planning and Root Canal during pregnancy. This knowledge about the safety of dental procedures during pregnancy should be addressed to improve service utilization and oral health promotion among pregnant mothers. Similarly, a study conducted among gynaecologists indicated a knowledge gap about the safety of some of dental procedures for pregnant mothers [39]. Local anesthetics lidocaine commonly used during dental care are safe, although, with some negligible adverse effect [40], some of these effects may include immune response-mediated allergic reactions and others that are unrelated to the immune response and these allergic responses are very rare in practice [41].

About the safety of obtaining dental radiographs in pregnant women, the majority 125 (82.2%) of the ANC providers were not aware that it is safe to obtain dental radiographs in pregnant women and use of dental radiographs, if necessary, after the first trimester does not pose any risk to the developing fetus [42], however, it is recommended to use protective lead aprons and thyroid collar to shield the sensitive areas. This kind of misconception could however be overcome among ANC providers by continuous services professional training and collaboration among different professionals within the different health facilities.

In our study, respondents who had no skills to advance OHP in ANC were less likely to adopt oral health promotion compared to those who said they had skills in advancing OHP for prenatal care. This calls for the need to incorporate prenatal oral care in health higher education training institutions of learning and continuous professional education for ANC providers in practice. Health training institutions have limited training for prenatal oral healthcare [43], adequate skills attained by health workers would enable them to perform desired tasks, more regular training and paying attention to discuss career development prospects would be of great value to advance their skills [44].

In addition, our study found that, having a good understanding between dentists and ANC providers contributed to adoption. A study conducted in Australia found no interprofessional collaboration and understanding between non-dentists and dentists [45]. An inter-professional collaboration education model for dental and medical providers are needed and the provision of an appropriate referral system for comprehensive clinical care of pregnant patients and accredited standards that encourage development and implementation [46, 47]. There is a need for collaboration between ANC providers with dentists to encourage all pregnant women to comply with the oral health professionals' recommendations regarding appropriate dental brushing techniques and the importance of dental visits [48]. However, Collaboration across organizational boundaries remains challenging due to power dynamics and trust that affect the strategic choices made by each health professional about whether to collaborate, with

whom and to what level [49]. These decisions directly affect inter-professional relationships trust and respect can be fostered through a mix of interventions aimed at building personal relationships and establishing agreed rules that govern collaborative care and that are perceived as fair to all professions. Some of the factors that facilitate the integration of oral health are interdisciplinary education and collaborative practices between dental and other healthcare professionals [8]. Adoption of oral promotion requires coordination and cooperation between ANC providers and dentists [50]. Understanding and interprofessional collaboration improve professional output, through the provision of clarity around the meaning and definition of professional roles [51].

In terms of management, respondents who said management was not good at influencing new practices among ANC providers were less likely to adopt oral health promotion compared to management being good at influencing new practices. Poor leadership were a key factor leading to providers' weak workplace trust and contributed to often-poor quality services, driving a perverse cycle of negative patient-provider relations [52]. For leaders to have lasting influence, staff members should be involved in health policy development and some of the key participatory roles in health policy development include mentoring, supporting and developing future policymakers [53].

In addition, the Ottawa charter emphasized that all sectors at all levels be involved in formulating policy and direct them to be aware of the health consequences of their decisions and to accept their responsibilities for health, health promotion policy requires the identification of obstacles to the adoption of healthy public policies and ways of removing them, thus health policies must focus on making it easy for beneficiaries to make healthier choices [54]. Involvement of staff members would help them on what is in the policy and also improve on policy dissemination and implementation at all levels, avoiding situations wherein a district, some oral health professionals have not even seen the national oral health policy until it was presented to them during the research work.

Moreover, our study found that there is a need to develop oral health promotion guidelines for pregnant mothers and according to [34], many countries have developed oral health guidelines for pregnant mothers to help them improve the oral health of both pregnant mothers and the babies and the main barriers for ANC providers in promoting oral health was lack of practice guidelines on oral health care during pregnancy. There is a need for local and national health policy agenda in Uganda to incorporate the prevention of oral diseases during pregnancy [23]. Burke-Litwin model noted that, when staff members feel that organizational policies and procedures are favourable to them, they are moted to perform the required actions, implying that appropriate formulation and implementation of policies motivate staff members to perform required actions.

## Study strength and limitations

Our study had several strengths, firstly, the large sample size covering three quarters of the health workers and all categories of cadres of health workers in the district, secondly, our study was the first in Uganda with a focus on the adoption of oral health promotion by antenatal care providers, we involved research assistants who were well trained to create rapport with respondents, thirdly, we trained research assistants on data collection procedures, maintaining confidentiality, ensure correctness and consistency of recorded data. Fourth, all our research tools were fully pretested and checked for accuracy and correctness, only filled questionnaires were included for analysis. However, our study could not be used to analyse behaviour or practices of the ANC providers over some time as it was a snapshot, limitation of confounding bias which was controlled by restricting participation to only those who had worked for at least six

months and above and participation to only ANC provider who had ever worked or are working at ANC department in order to arrive at meaningful and conclusive results.

## Conclusion

Adoption of OHP was low. Factors influencing the adoption of oral health promotion were; age of respondents, the number of years at work, good understanding between dentists and ANC providers, and management being good at influencing new practices contributed to an increased level in the adoption of oral health promotion. However, knowledge gap, poor dissemination of National oral health policy and lack of prenatal oral health guidelines were some of the barriers to an increased level of adoption by antenatal care providers. We, therefore, recommend that the current NOHP should be reviewed, develop prenatal OHC guidelines, enhance the capacity of ANC providers through training, collaboration with dentists and launch official adoption of OHP.

## Acknowledgments

We are grateful to Uganda Martyrs University for awarding the primary author with a master degree in Public Health–Health Promotion. We thank International Health Sciences University (Currently, Clarke International University) research ethics committee for according us a waiver for our protocol. we appreciate all our research assistants for the support rendered during the process of data collection.

## Author Contributions

**Conceptualization:** Patrick Madrama Lulu, Miisa Nanyingi.

**Data curation:** Patrick Madrama Lulu, Miisa Nanyingi.

**Formal analysis:** Patrick Madrama Lulu.

**Investigation:** Patrick Madrama Lulu.

**Methodology:** Patrick Madrama Lulu.

**Project administration:** Patrick Madrama Lulu.

**Resources:** Patrick Madrama Lulu.

**Software:** Patrick Madrama Lulu.

**Supervision:** Miisa Nanyingi.

**Validation:** Patrick Madrama Lulu.

**Visualization:** Patrick Madrama Lulu.

**Writing – original draft:** Patrick Madrama Lulu.

**Writing – review & editing:** Patrick Madrama Lulu, Miisa Nanyingi.

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
