## [Decision Letter · Decision Letter 0]

19 Oct 2022

PONE-D-22-18056Factors influencing adoption of oral health promotion by antenatal care providers in Moyo district, North-Western UgandaPLOS ONE

Dear Dr. Lulu,

Thank you for submitting your manuscript to PLOS ONE. After careful consideration, we feel that it has merit but does not fully meet PLOS ONE’s publication criteria as it currently stands. Therefore, we invite you to submit a revised version of the manuscript that addresses the points raised during the review process.

Please read carefully the comments in the attached PDF file/s and address each question from the reviewer.

Please submit your revised manuscript by 19th November. If you will need more time than this to complete your revisions, please reply to this message or contact the journal office at plosone@plos.org. Please include the following items when submitting your revised manuscript:A rebuttal letter that responds to each point raised by the academic editor and reviewer(s). You should upload this letter as a separate file labeled 'Response to Reviewers'.A marked-up copy of your manuscript that highlights changes made to the original version. You should upload this as a separate file labeled 'Revised Manuscript with Track Changes'.An unmarked version of your revised paper without tracked changes. You should upload this as a separate file labeled 'Manuscript'.

We look forward to receiving your revised manuscript.

Kind regards,

Muhammad Farooq Umer, BDS, MSPH, FRSPH, PhD Epidemiology and Health Stat

Academic Editor

PLOS ONE

Journal Requirements:

2.  Please update your submission to use the PLOS LaTeX template. The template and more information on our requirements for LaTeX submissions can be found at http://journals.plos.org/plosone/s/latex

Reviewers' comments:

Reviewer's Responses to Questions

**Comments to the Author**

1. Is the manuscript technically sound, and do the data support the conclusions?

Reviewer #1: Yes

Reviewer #2: Partly

2. Has the statistical analysis been performed appropriately and rigorously? 

Reviewer #1: Yes

Reviewer #2: Yes

3. Have the authors made all data underlying the findings in their manuscript fully available?

Reviewer #1: Yes

Reviewer #2: Yes

4. Is the manuscript presented in an intelligible fashion and written in standard English?

Reviewer #1: Yes

Reviewer #2: Yes

5. Review Comments to the Author

Reviewer #1: Dear Authors,

I appreciated the entirety of your research article, it comprehends several aspects that are frequently assessed for different diseases in the oral cavity during pregnancy and antenatal care. Overall details shared in the article are adequately explained via appropriate tables ad figures. The present manuscript is well written, organized and the topic is interesting for the leadership of oral diseases. Minor changes are recommended to improve the manuscript of this article.

Reviewer #2: this work is an important contribution to the oral health promotion

the work can be very much developed by rearranging the manuscript

the results will be better if presenting the quantitative separately and then the qualitative part

some parts of the methods need corrections and additions

some figures in the tables need revision

please check the recommendations in the pdf format

6. PLOS authors have the option to publish the peer review history of their article (what does this mean?). If published, this will include your full peer review and any attached files.

Reviewer #1: **Yes: **Dr. Afsheen Mansoor

Associate Professor

ACMED, MSHM, PhD.

Gold Medalist

School of Dentistry, Shaheed Zulfiqar Ali Bhutto Medical University (SZABMU) Islamabad

Pakistan

Ph # 92-321-5879166

drafsheen@szabmu.edu.pk and

drafsheenqamar@gmail.com

Reviewer #2: **Yes: **Elwalid Fadul Nasir

---

## [Author Response · Author response to Decision Letter 0]

17 Dec 2022

Patrick Madrama Lulu,

 Clarke International University,

 P.0. Box 7782, Kampala- Uganda

 14th .11.2022 

Dear Emily Chenette,

Editor in Chief, Plos One Journal

Ref: Response to Reviewers for “Factors influencing adoption of oral health promotion by antenatal care providers in Moyo district, North-Western Uganda.”

Thank you for taking time to review our manuscript that was submitted to you, thank you for the comments. Below are the responses to the reviewers.

Section Reviewers’ Comment Responses 

Abstract you may need to insert the OR and CIF for the first 2 factors as well OR and CIF have been inserted for the first 2 factors.

 Delete a a, have been deleted from aOR

Background 

 This sentence better to come after the second sentence in the introduction as it is continuation regarding oral health followed by the sentence starting with Antenatal care 

The suggested sentence has been inserted after the second sentence in the introduction as suggested by the reviewers.

Study Population and settings

 Exclusion criteria is to be revised!

as a general principle exclusion is to exclude the already included (fulfilling the inclusion criteria) but with some characteristics they are to be excluded. 

Exclusion criteria has been revised following general principles.

Study Variables

 What was the logic behind applying these cut-off points?

I would suggest to use the mean score as the cut-off point! The researchers have now clearly indicated the values for mean scores for each cut off point.

 what was the cut-off point in these two variables The cut-off point for these two variables have now been clearly indicated.

Study Design

 Could you please elaborate on the equation by adding the numbers used in the equation e.g. population size 

The formula for calculating the sample size has been elaborated as suggested by the reviewers and population size indicated.

Data Analysis 

 what was the level of significance used? 

This has been adjusted.

Results 

 This is not shown the table 1 Has been adjusted as required.

 what is the expected level and the actual level measured? It was an error, expected levels have been removed.

 This table would be better presented in two columns the second containing both frequency and parentage! This table 1 has been adjusted as suggested by reviewers.

 This should be 69.7

the total adds to 100.2% This value has been adjusted in table 1, as required.

 This should be 54% This has as well been adjusted as required.

 You need to mention the OR and CI OR and CI have been added.

 ONLY this age-group was significant others were not

check the OR and CI AND p-values as well! This has been adjusted as required.

 this comparison is wrong!!!

the reference group is 31-40 years’ experience both groups are compared to the last group of experience (they are not significant look at te CI) REVISE This has been revised.

 This comparison is wrong the reference group is HC IV This has been adjusted.

 Wrong interpretation of the UOR!!

REVISE The interpretation has been revised.

 compared to those had skills!! This comparison has been adjusted.

 Mention the livelihood between the two groups UOR three times, eight times and twice The likelihood between the two groups have been revised.

 This sentence belong to the discussion than the results section The sentence has been removed from results section.

 These results depicted in figure 3 NOT one This has been adjusted as required.

 These results are depicted in figure 1. Has been adjusted as required.

 What was the CI This has been revised in table 3.

Strength and Limitations

Data analyses does not control for any bias! 

This has been revised.

References 

 Any abbreviation for this journal! 

Abbreviation for the journals have been indicated as required.

Yours Sincerely

Patrick Madrama Lulu

Phone: +256 782 98 04 51

Email:Lulu.madrama@gmail.com

Consultant Research Ethics Committee/Lecturer, Clarke International University

---

## [Decision Letter · Decision Letter 1]

7 Feb 2023

PONE-D-22-18056R1Factors influencing adoption of oral health promotion by antenatal care providers in Moyo district, North-Western UgandaPLOS ONE

Dear Dr. Lulu,

Thank you for submitting your manuscript to PLOS ONE. After careful consideration, we feel that it has merit but does not fully meet PLOS ONE’s publication criteria as it currently stands. Therefore, we invite you to submit a revised version of the manuscript that addresses the points raised during the review process.

ACADEMIC EDITOR:  Please address the comments from one of the reviewers who has asked to review English language and back your analysis with more graph(s) where necessary.Please complete the revision within next 10 days so that editorial process for this research may be completed soon as possible.

We look forward to receiving your revised manuscript.

Kind regards,

Muhammad Farooq Umer,

Academic Editor

PLOS ONE

Journal Requirements:

Reviewers' comments:

Reviewer's Responses to Questions

**Comments to the Author**

1. If the authors have adequately addressed your comments raised in a previous round of review and you feel that this manuscript is now acceptable for publication, you may indicate that here to bypass the “Comments to the Author” section, enter your conflict of interest statement in the “Confidential to Editor” section, and submit your "Accept" recommendation.

Reviewer #3: (No Response)

Reviewer #4: All comments have been addressed

2. Is the manuscript technically sound, and do the data support the conclusions?

Reviewer #3: Yes

Reviewer #4: Yes

3. Has the statistical analysis been performed appropriately and rigorously? 

Reviewer #3: Yes

Reviewer #4: Yes

4. Have the authors made all data underlying the findings in their manuscript fully available?

Reviewer #3: No

Reviewer #4: Yes

5. Is the manuscript presented in an intelligible fashion and written in standard English?

Reviewer #3: No

Reviewer #4: Yes

6. Review Comments to the Author

Reviewer #3: Please add more graphs to support your results

the result section needs to be re organised.

minor language changes are advised. long continous sentences can be made short and clear

Reviewer #4: The topic is an important issue ,research is good.Author has well explained the methodology and embedded all the amendments highlighted by the first reviewer

7. PLOS authors have the option to publish the peer review history of their article (what does this mean?). If published, this will include your full peer review and any attached files.

Reviewer #3: No

Reviewer #4: No

---

## [Author Response · Author response to Decision Letter 1]

28 Mar 2023

Patrick Madrama Lulu,

 Clarke International University,

 P.0. Box 7782, Kampala- Uganda

 25th .02.2023 

Dear Emily Chenette,

Editor in Chief, Plos One Journal

Ref: Response to Reviewers for “Factors influencing adoption of oral health promotion by antenatal care providers in Moyo district, North-Western Uganda.”

Thank you for taking time to review our manuscript that was submitted to you, thank you for the comments. Below are the responses to the reviewers.

Section Reviewers’ Comment Responses 

Materials and Methods Please elaborate about your exclusion criteria We were looking at the fact that Medical Doctors, Nurses, Clinical Officers etc who did not work at ANC department did not have much opportunity to care for pregnant mothers, therefore, did not qualify for the study. 

ANC providers who had worked within the district less than six months may not be able to give authentic responses related to the study within the area of the study.

Study Variables Items used to measure adoption: list of items used??? Items have been listed.

 please mention the list of practice items that was used Items have been listed.

 List of items used to measure Knowledge Items have been listed.

 List of items used to measure attitude Items have been listed.

 List of items used to measure capacity of health facilities. Items have been listed.

Results 

 please reformat this table, title for the column is missing in Table 1, has unlabelled column This has been formated and deleted in line comments from previous first review.

 Clinical Officers - 2013.2 in table 1 This has been corrected 20(13.2)

 what was your basis for this conclusion??

How was the adoption measured? Adoption was measured as described under the variables in page 4 of this manuscript.

General Please address the comments from one of the reviewers who has asked to review English language and back your analysis with more graph(s) where necessary. Review of English language have been done.

Our manuscript has the necessary supporting figures and tables in line with plos one guidelines.

Yours Sincerely

Patrick Madrama Lulu

Phone: +256 782 98 04 51

Email:Lulu.madrama@gmail.com

Consultant Research Ethics Committee/Lecturer, Clarke International University

---

## [Editor Report · Decision Letter 2]

10 Apr 2023

Factors influencing adoption of oral health promotion by antenatal care providers in Moyo district, North-Western Uganda

PONE-D-22-18056R2

Dear Dr. Lulu,

We’re pleased to inform you that your manuscript has been judged scientifically suitable for publication and will be formally accepted for publication once it meets all outstanding technical requirements.

Kind regards,

Muhammad Farooq Umer

Academic Editor

PLOS ONE
---

## [Editor Report · Acceptance letter]

13 Apr 2023

PONE-D-22-18056R2 

Factors influencing adoption of oral health promotion by antenatal care providers in Moyo district, North-Western Uganda 

Dear Dr. Lulu:

I'm pleased to inform you that your manuscript has been deemed suitable for publication in PLOS ONE. Congratulations! Your manuscript is now with our production department. 

Kind regards, 

on behalf of

Dr. Muhammad Farooq Umer 

Guest Editor

PLOS ONE